# Reward Augmented Maximum Likelihood
# for Neural Structured Prediction

**Mohammad Norouzi**       **Samy Bengio**       **Zhifeng Chen**       **Navdeep Jaitly**
**Mike Schuster**       **Yonghui Wu**       **Dale Schuurmans**

{mnorouzi, bengio, zhifengc, ndjaitly}@google.com
{schuster, yonghui, schuurmans}@google.com

Google Brain

## Abstract

A key problem in structured output prediction is direct optimization of the task reward function that matters for test evaluation. This paper presents a simple and computationally efficient approach to incorporate task reward into a maximum likelihood framework. By establishing a link between the log-likelihood and expected reward objectives, we show that an optimal regularized expected reward is achieved when the conditional distribution of the outputs given the inputs is proportional to their exponentiated scaled rewards. Accordingly, we present a framework to smooth the predictive probability of the outputs using their corresponding rewards. We optimize the conditional log-probability of augmented outputs that are sampled proportionally to their exponentiated scaled rewards. Experiments on neural sequence to sequence models for speech recognition and machine translation show notable improvements over a maximum likelihood baseline by using reward augmented maximum likelihood (RML), where the rewards are defined as the negative edit distance between the outputs and the ground truth labels.

## 1 Introduction

Structured output prediction is ubiquitous in machine learning. Recent advances in natural language processing, machine translation, and speech recognition hinge on the development of better discriminative models for structured outputs and sequences. The foundations of learning structured output models were established by the seminal work on conditional random fields (CRFs) [17] and structured large margin methods [32], which demonstrate how generalization performance can be significantly improved when one considers the joint effects of the predictions across multiple output components. These models have evolved into their deep neural counterparts [29, 1] through the use of recurrent neural networks (RNN) with LSTM [13] cells and attention mechanisms [2].

A key problem in structured output prediction has always been to enable direct optimization of the task reward (loss) used for test evaluation. For example, in machine translation one seeks better BLEU scores, and in speech recognition better word error rates. Not surprisingly, almost all task reward metrics are not differentiable, hence hard to optimize. Neural sequence models (*e.g.* [29, 2]) optimize conditional log-likelihood, *i.e.* the conditional log-probability of the ground truth outputs given corresponding inputs. These models do not explicitly consider the task reward during training, hoping that conditional log-likelihood serves as a good surrogate for the task reward. Such methods make no distinction between alternative incorrect outputs: log-probability is only measured on the ground truth input-output pairs, and all alternative outputs are equally penalized through normalization, whether near or far from the ground truth target. We believe one can improve upon maximum likelihood (ML) sequence models if the difference in the rewards of alternative outputs is taken into account.

Standard ML training, despite its limitations, has enabled the training of deep RNN models, leading to revolutionary advances in machine translation [29, 2, 21] and speech recognition [5–7]. A key property

of ML training for locally normalized RNN models is that the objective function factorizes into individual loss terms, which could be *efficiently* optimized using stochastic gradient descend (SGD). This training procedure does not require any form of inference or sampling from the model during training, leading to computational efficiency and ease to implementation. By contrast, almost all alternative formulations for training structure prediction models require some form of inference or sampling from the model at training time which slows down training, especially for deep RNNs (*e.g.* see large margin, search-based [8, 39], and expected risk optimization methods).

Our work is inspired by the use of reinforcement learning (RL) algorithms, such as policy gradient [37], to optimize expected task reward [25]. Even though expected task reward seems like a natural objective, direct policy optimization faces significant challenges: unlike ML, a stochastic gradient given a mini-batch of training examples is extremely noisy and has a high variance; gradients need to be estimated via sampling from the model, which is a non-stationary distribution; the reward is often sparse in a high-dimensional output space, which makes it difficult to find any high value predictions, preventing learning from getting off the ground; and, finally, maximizing reward does not explicitly consider the supervised labels, which seems inefficient. In fact, all previous attempts at direct policy optimization for structured output prediction have started by bootstrapping from a previously trained ML solution [25, 27], using several heuristics and tricks to make learning stable.

This paper presents a new approach to task reward optimization that combines the computational efficiency and simplicity of ML with the conceptual advantages of expected reward maximization. Our algorithm called *reward augmented maximum likelihood (RML)* simply adds a sampling step on top of the typical likelihood objective. Instead of optimizing conditional log-likelihood on training input-output pairs, given each training input, we first sample an output proportionally to its exponentiated scaled reward. Then, we optimize log-likelihood on such auxiliary output samples given corresponding inputs. When the reward for an output is defined as its similarity to a ground truth output, then the output sampling distribution is peaked at the ground truth output, and its concentration is controlled by a temperature hyper-parameter.

Our theoretical analysis shows that the RML and regularized expected reward objectives optimize a KL divergence between the exponentiated reward and model distributions, but in opposite directions. Further, we show that at non-zero temperatures, the gap between the two criteria can be expressed by a difference of variances measured on interpolating distributions. This observation reveals how entropy regularized expected reward can be estimated by sampling from exponentiated scaled rewards, rather than sampling from the model distribution.

Remarkably, we find that the RML approach achieves significantly improved results over state of the art maximum likelihood RNNs. We show consistent improvement on both speech recognition (TIMIT dataset) and machine translation (WMT'14 dataset), where output sequences are sampled according to their edit distance to the ground truth outputs. Surprisingly, we find that the best performance is achieved with output sampling distributions that shift a lot of the weight away from the ground truth outputs. In fact, in our experiments, the training algorithm rarely sees the original unperturbed outputs. Our results give further evidence that models trained with imperfect outputs and their reward values can improve upon models that are only exposed to a single ground truth output per input [12, 20].

## 2 Reward augmented maximum likelihood

Given a dataset of input-output pairs, $\mathcal{D} \equiv \{(\mathbf{x}^{(i)}, \mathbf{y}^{*(i)})\}_{i=1}^{N}$, structured output models learn a parametric score function $p_\theta(\mathbf{y} \mid \mathbf{x})$, which scores different output hypotheses, $\mathbf{y} \in \mathcal{Y}$. We assume that the set of possible output, $\mathcal{Y}$ is finite, *e.g.* English sentences up to a maximum length. In a probabilistic model, the score function is normalized, while in a large-margin model the score may not be normalized. In either case, once the score function is learned, given an input $\mathbf{x}$, the model predicts an output $\widehat{\mathbf{y}}$ achieving maximal score,

$$\widehat{\mathbf{y}}(\mathbf{x}) = \underset{\mathbf{y}}{\mathrm{argmax}}\, p_\theta(\mathbf{y} \mid \mathbf{x}) . \tag{1}$$

If this optimization is intractable, approximate inference (*e.g.* beam search) is used. We use a reward function $r(\mathbf{y}, \mathbf{y}^*)$ to evaluate different proposed outputs against ground-truth outputs. Given a test dataset $\mathcal{D}'$, one computes $\sum_{(\mathbf{x}, \mathbf{y}^*) \in \mathcal{D}'} r(\widehat{\mathbf{y}}(\mathbf{x}), \mathbf{y}^*)$ as a measure of empirical reward. Since models with larger empirical reward are preferred, ideally one hopes to maximize empirical reward during training.

However, since empirical reward is not amenable to numerical optimization, one often considers optimizing alternative differentiable objectives. The maximum likelihood (ML) framework tries to minimize negative log-likelihood of the parameters given the data,

$$\mathcal{L}_{\mathrm{ML}}(\boldsymbol{\theta}; \mathcal{D}) = \sum_{(\mathbf{x}, \mathbf{y}^*) \in \mathcal{D}} -\log p_\theta(\mathbf{y}^* \mid \mathbf{x}) \,. \tag{2}$$

Minimizing this objective increases the conditional probability of the target outputs, $\log p_\theta(\mathbf{y}^* \mid \mathbf{x})$, while decreasing the conditional probability of alternative incorrect outputs. According to this objective, all negative outputs are equally wrong, and none is preferred over the others.

By contrast, reinforcement learning (RL) advocates optimizing expected reward (with a maximum entropy regularizer [38]), which is formulated as minimization of the following objective,

$$\mathcal{L}_{\mathrm{RL}}(\boldsymbol{\theta}; \tau, \mathcal{D}) = \sum_{(\mathbf{x}, \mathbf{y}^*) \in \mathcal{D}} \left\{ -\tau \mathbb{H} \left( p_\theta(\mathbf{y} \mid \mathbf{x}) \right) - \sum_{\mathbf{y} \in \mathcal{Y}} p_\theta(\mathbf{y} \mid \mathbf{x}) \, r(\mathbf{y}, \mathbf{y}^*) \right\}, \tag{3}$$

where $r(\mathbf{y}, \mathbf{y}^*)$ denotes the reward function, *e.g.* negative edit distance or BLEU score, $\tau$ controls the degree of regularization, and $\mathbb{H}(p)$ is the entropy of a distribution $p$, *i.e.* $\mathbb{H}(p(\mathbf{y})) = -\sum_{\mathbf{y} \in \mathcal{Y}} p(\mathbf{y}) \log p(\mathbf{y})$. It is well-known that optimizing $\mathcal{L}_{\mathrm{RL}}(\boldsymbol{\theta}; \tau)$ using SGD is challenging because of the large variance of the gradients. Below we describe how ML and RL objectives are related, and propose a hybrid between the two that combines their benefits for supervised learning.

Let us define a distribution in the output space, termed the *exponentiated payoff distribution*, that is central in linking ML and RL objectives:

$$q(\mathbf{y} \mid \mathbf{y}^*; \tau) = \frac{1}{Z(\mathbf{y}^*, \tau)} \exp \left\{ r(\mathbf{y}, \mathbf{y}^*)/\tau \right\}, \tag{4}$$

where $Z(\mathbf{y}^*, \tau) = \sum_{\mathbf{y} \in \mathcal{Y}} \exp \left\{ r(\mathbf{y}, \mathbf{y}^*)/\tau \right\}$. One can verify that the global minimum of $\mathcal{L}_{\mathrm{RL}}(\boldsymbol{\theta}; \tau)$, *i.e.* the optimal regularized expected reward, is achieved when the model distribution matches the exponentiated payoff distribution, *i.e.* $p_\theta(\mathbf{y} \mid \mathbf{x}) = q(\mathbf{y} \mid \mathbf{y}^*; \tau)$. To see this, we re-express the objective function in (3) in terms of a KL divergence between $p_\theta(\mathbf{y} \mid \mathbf{x})$ and $q(\mathbf{y} \mid \mathbf{y}^*; \tau)$,

$$\sum_{(\mathbf{x}, \mathbf{y}^*) \in \mathcal{D}} D_{\mathrm{KL}} \left( p_\theta(\mathbf{y} \mid \mathbf{x}) \parallel q(\mathbf{y} \mid \mathbf{y}^*; \tau) \right) = \frac{1}{\tau} \mathcal{L}_{\mathrm{RL}}(\boldsymbol{\theta}; \tau) + \mathrm{const} \,, \tag{5}$$

where the constant $\mathrm{const}$ on the RHS is $\sum_{(\mathbf{x}, \mathbf{y}^*) \in \mathcal{D}} \log Z(\mathbf{y}^*, \tau)$. Thus, the minimum of $D_{\mathrm{KL}} \left( p_\theta \parallel q \right)$ and $\mathcal{L}_{\mathrm{RL}}$ is achieved when $p_\theta = q$. At $\tau = 0$, when there is no entropy regularization, the optimal $p_\theta$ is a delta distribution, $p_\theta(\mathbf{y} \mid \mathbf{x}) = \delta(\mathbf{y} \mid \mathbf{y}^*)$, where $\delta(\mathbf{y} \mid \mathbf{y}^*) = 1$ at $\mathbf{y} = \mathbf{y}^*$ and $0$ at $\mathbf{y} \neq \mathbf{y}^*$. Note that $\delta(\mathbf{y} \mid \mathbf{y}^*)$ is equivalent to the exponentiated payoff distribution in the limit as $\tau \to 0$.

Returning to the log-likelihood objective, one can verify that (2) is equivalent to a KL divergence in the opposite direction between a delta distribution $\delta(\mathbf{y} \mid \mathbf{y}^*)$ and the model distribution $p_\theta(\mathbf{y} \mid \mathbf{x})$,

$$\sum_{(\mathbf{x}, \mathbf{y}^*) \in \mathcal{D}} D_{\mathrm{KL}} \left( \delta(\mathbf{y} \mid \mathbf{y}^*) \parallel p_\theta(\mathbf{y} \mid \mathbf{x}) \right) = \mathcal{L}_{\mathrm{ML}}(\boldsymbol{\theta}) \,. \tag{6}$$

There is no constant on the RHS, as the entropy of a delta distribution is zero, *i.e.* $\mathbb{H} \left( \delta(\mathbf{y} \mid \mathbf{y}^*) \right) = 0$.

We propose a method called *reward-augmented maximum likelihood (RML)*, which generalizes ML by allowing a non-zero temperature parameter in the exponentiated payoff distribution, while still optimizing the KL divergence in the ML direction. The RML objective function takes the form,

$$\mathcal{L}_{\mathrm{RML}}(\boldsymbol{\theta}; \tau, \mathcal{D}) = \sum_{(\mathbf{x}, \mathbf{y}^*) \in \mathcal{D}} \left\{ -\sum_{\mathbf{y} \in \mathcal{Y}} q(\mathbf{y} \mid \mathbf{y}^*; \tau) \log p_\theta(\mathbf{y} \mid \mathbf{x}) \right\}, \tag{7}$$

which can be re-expressed in terms of a KL divergence as follows,

$$\sum_{(\mathbf{x}, \mathbf{y}^*) \in \mathcal{D}} D_{\mathrm{KL}} \left( q(\mathbf{y} \mid \mathbf{y}^*; \tau) \parallel p_\theta(\mathbf{y} \mid \mathbf{x}) \right) = \mathcal{L}_{\mathrm{RML}}(\boldsymbol{\theta}; \tau) + \mathrm{const} \,, \tag{8}$$

where the constant $\mathrm{const}$ is $-\sum_{(\mathbf{x}, \mathbf{y}^*) \in \mathcal{D}} \mathbb{H} \left( q(\mathbf{y} \mid \mathbf{y}^*, \tau) \right)$. Note that the temperature parameter, $\tau \geq 0$, serves as a hyper-parameter that controls the smoothness of the optimal distribution around

correct targets by taking into account the reward function in the output space. The objective functions $\mathcal{L}_{\mathrm{RL}}(\boldsymbol{\theta}; \tau)$ and $\mathcal{L}_{\mathrm{RML}}(\boldsymbol{\theta}; \tau)$, have the same global optimum of $p_\theta$, but they optimize a KL divergence in opposite directions. We characterize the difference between these two objectives below, showing that they are equivalent up to their first order Taylor approximations. For optimization convenience, we focus on minimizing $\mathcal{L}_{\mathrm{RML}}(\boldsymbol{\theta}; \tau)$ to achieve a good solution for $\mathcal{L}_{\mathrm{RL}}(\boldsymbol{\theta}; \tau)$.

## 2.1 Optimization

Optimizing the reward augmented maximum likelihood (RML) objective, $\mathcal{L}_{\mathrm{RML}}(\boldsymbol{\theta}; \tau)$, is straightforward if one can draw unbiased samples from $q(\mathbf{y} \mid \mathbf{y}^*; \tau)$. We can express the gradient of $\mathcal{L}_{\mathrm{RML}}$ in terms of an expectation over samples from $q(\mathbf{y} \mid \mathbf{y}^*; \tau)$,

$$\nabla_{\boldsymbol{\theta}} \mathcal{L}_{\mathrm{RML}}(\boldsymbol{\theta}; \tau) \ = \ \mathbb{E}_{q(\mathbf{y}|\mathbf{y}^*;\tau)} \left[ - \nabla_{\boldsymbol{\theta}} \log p_\theta(\mathbf{y} \mid \mathbf{x}) \right] . \tag{9}$$

Thus, to estimate $\nabla_{\boldsymbol{\theta}} \mathcal{L}_{\mathrm{RML}}(\boldsymbol{\theta}; \tau)$ given a mini-batch of examples for SGD, one draws $\mathbf{y}$ samples given mini-batch $\mathbf{y}^*$'s and then optimizes log-likelihood on such samples by following the mean gradient. At a temperature $\tau = 0$, this reduces to always sampling $\mathbf{y}^*$, hence ML training with no sampling.

By contrast, the gradient of $\mathcal{L}_{\mathrm{RL}}(\boldsymbol{\theta}; \tau)$, based on likelihood ratio methods, takes the form,

$$\nabla_{\boldsymbol{\theta}} \mathcal{L}_{\mathrm{RL}}(\boldsymbol{\theta}; \tau) \ = \ \mathbb{E}_{p_\theta(\mathbf{y}|\mathbf{x})} \left[ - \nabla_{\boldsymbol{\theta}} \log p_\theta(\mathbf{y} \mid \mathbf{x}) \cdot r(\mathbf{y}, \mathbf{y}^*) \right] . \tag{10}$$

There are several critical differences between (9) and (10) that make SGD optimization of $\mathcal{L}_{\mathrm{RML}}(\boldsymbol{\theta}; \tau)$ more desirable. First, in (9), one has to sample from a stationary distribution, the so called exponentiated payoff distribution, whereas in (10) one has to sample from the model distribution as it is evolving. Not only does sampling from the model potentially slow down training, one also needs to employ several tricks to get a better estimate of the gradient of $\mathcal{L}_{\mathrm{RL}}$ [25]. A body of literature in reinforcement learning focuses on reducing the variance of (10) by using sophisticated techniques such as *actor-critique* methods [30, 9]. Further, the reward is often sparse in a high-dimensional output space, which makes finding any reasonable prediction challenging when (10) is used to refine a randomly initialized model. Thus, smart model initialization is needed. By contrast, we initialize the models randomly and refine them using (9).

## 2.2 Sampling from the exponentiated payoff distribution

To compute the gradient of the model using the RML approach, one needs to sample auxiliary outputs from the exponentiated payoff distribution, $q(\mathbf{y} \mid \mathbf{y}^*; \tau)$. This sampling is the price that we have to pay to learn with rewards. One should contrast this with loss-augmented inference in structured large margin methods, and sampling from the model in RL. We believe sampling outputs proportional to exponentiated rewards is more efficient and effective in many cases.

Experiments in this paper use reward values defined by either negative Hamming distance or negative edit distance. We sample from $q(\mathbf{y} \mid \mathbf{y}^*; \tau)$ by stratified sampling, where we first select a particular distance, and then sample an output with that distance value. Here we focus on edit distance sampling, as Hamming distance sampling is a simpler special case. Given a sentence $\mathbf{y}^*$ of length $m$, we count the number of sentences within an edit distance $e$, where $e \in \{0, \ldots, 2m\}$. Then, we reweight the counts by $\exp\{-e/\tau\}$ and normalize. Let $c(e, m)$ denote the number of sentences at an edit distance $e$ from a sentence of length $m$. First, note that a deletion can be thought as a substitution with a nil token. This works out nicely because given a vocabulary of length $v$, for each insertion we have $v$ options, and for each substitution we have $v - 1$ options, but including the nil token, there are $v$ options for substitutions too. When $e = 1$, there are $m$ possible substitutions and $m + 1$ insertions. Hence, in total there are $(2m + 1)v$ sentences at an edit distance of 1. Note, that exact computation of $c(e, m)$ is difficult if we consider all edge cases, for example when there are repetitive words in $\mathbf{y}^*$, but ignoring such edge cases we can come up with approximate counts that are reliable for sampling. When $e > 1$, we estimate $c(e, m)$ by

$$c(e, m) = \sum_{s=0}^{m} \binom{m}{s} \binom{m + e - 2s}{e - s} v^e , \tag{11}$$

where $s$ enumerates over the number of substitutions. Once $s$ tokens are substituted, then those $s$ positions lose their significance, and the insertions before and after such tokens could be merged. Hence, given $s$ substitutions, there are really $m - s$ reference positions for $e - s$ possible insertions. Finally, one can sample according to BLEU score or other sequence metrics by importance sampling where the proposal distribution could be edit distance sampling above.

## 3 RML analysis

In the RML framework, we find the model parameters by minimizing the objective (7) instead of optimizing the RL objective, *i.e.* regularized expected reward in (3). The difference lies in minimizing $D_{\mathrm{KL}}\left(q(\mathbf{y} \mid \mathbf{y}^*; \tau) \parallel p_\theta(\mathbf{y} \mid \mathbf{x})\right)$ instead of $D_{\mathrm{KL}}\left(p_\theta(\mathbf{y} \mid \mathbf{x}) \parallel q(\mathbf{y} \mid \mathbf{y}^*; \tau)\right)$. For convenience, let's refer to $q(\mathbf{y} \mid \mathbf{y}^*; \tau)$ as $q$, and $p_\theta(\mathbf{y} \mid \mathbf{x})$ as $p$. Here, we characterize the difference between the two divergences, $D_{\mathrm{KL}}\left(q \parallel p\right) - D_{\mathrm{KL}}\left(p \parallel q\right)$, and use this analysis to motivate the RML approach.

We will initially consider the KL divergence in its more general form as a Bregman divergence, which will make some of the key properties clearer. A Bregman divergence is defined by a strictly convex, differentiable, closed potential function $F : \mathcal{F} \to \mathbb{R}$ [3]. Given $F$ and two points $p, q \in \mathcal{F}$, the corresponding Bregman divergence $D_F : \mathcal{F} \times \mathcal{F} \to \mathbb{R}^+$ is defined by

$$D_F\left(p \parallel q\right) = F\left(p\right) - F\left(q\right) - \left(p - q\right)^{\mathsf{T}} \nabla F\left(q\right) , \tag{12}$$

the difference between the strictly convex potential at $p$ and its first order Taylor approximation expanded about $q$. Clearly this definition is not symmetric between $p$ and $q$. By the strict convexity of $F$ it follows that $D_F\left(p \parallel q\right) \geq 0$ with $D_F\left(p \parallel q\right) = 0$ if and only if $p = q$. To characterize the difference between opposite Bregman divergences, we provide a simple result that relates the two directions under suitable conditions. Let $H_F$ denote the Hessian of $F$.

**Proposition 1.** *For any twice differentiable strictly convex closed potential $F$, and $p, q \in \mathrm{int}(\mathcal{F})$:*

$$D_F\left(q \parallel p\right) = D_F\left(p \parallel q\right) + \tfrac{1}{4}(p - q)^{\mathsf{T}}\bigl(H_F(a) - H_F(b)\bigr)(p - q) \tag{13}$$

*for some $a = (1 - \alpha)p + \alpha q$, $(0 \leq \alpha \leq \tfrac{1}{2})$, $b = (1 - \beta)q + \beta p$, $(0 \leq \beta \leq \tfrac{1}{2})$. (see supp. material)*

For probability vectors $p, q \in \Delta^{|\mathcal{Y}|}$ and a potential $F\left(p\right) = -\tau \mathbb{H}\left(p\right)$, $D_F\left(p \parallel q\right) = \tau D_{\mathrm{KL}}\left(p \parallel q\right)$. Let $f^* : \mathbb{R}^{|\mathcal{Y}|} \to \Delta^{|\mathcal{Y}|}$ denote a normalized exponential operator that takes a real-valued logit vector and turns it into a probability vector. Let $r$ and $s$ denote real-valued logit vectors such that $q = f^*(r/\tau)$ and $p = f^*(s/\tau)$. Below, we characterize the gap between $D_{\mathrm{KL}}\left(p(y) \parallel q(y)\right)$ and $D_{\mathrm{KL}}\left(q(y) \parallel p(y)\right)$ in terms of the difference between $s(y)$ and $r(y)$.

**Proposition 2.** *The KL divergence between $p$ and $q$ in two directions can be expressed as,*

$$\begin{aligned} D_{\mathrm{KL}}\left(p \parallel q\right) &= D_{\mathrm{KL}}\left(q \parallel p\right) + \tfrac{1}{4\tau^2}\mathrm{Var}_{y \sim f^*(a/\tau)}\left[s(y) - r(y)\right] - \tfrac{1}{4\tau^2}\mathrm{Var}_{y \sim f^*(b/\tau)}\left[s(y) - r(y)\right] \\ &< D_{\mathrm{KL}}\left(q \parallel p\right) + \tfrac{1}{\tau^2}\left\|s - r\right\|_2^2, \end{aligned}$$

*for some $a = (1 - \alpha)s + \alpha r$, $(0 \leq \alpha \leq \tfrac{1}{2})$, $b = (1 - \beta)r + \beta s$, $(0 \leq \beta \leq \tfrac{1}{2})$. (see supp. material)*

Given Proposition 2, one can relate the two objectives, $\mathcal{L}_{\mathrm{RL}}(\boldsymbol{\theta}; \tau)$ (5) and $\mathcal{L}_{\mathrm{RML}}(\boldsymbol{\theta}; \tau)$ (8), by

$$\mathcal{L}_{\mathrm{RL}} = \tau \mathcal{L}_{\mathrm{RML}} + \tfrac{1}{4\tau} \sum_{(\mathbf{x}, \mathbf{y}^*) \in \mathcal{D}} \left\{ \mathrm{Var}_{\mathbf{y} \sim f^*(a/\tau)}\left[s(\mathbf{y}) - r(\mathbf{y})\right] - \mathrm{Var}_{\mathbf{y} \sim f^*(b/\tau)}\left[s(\mathbf{y}) - r(\mathbf{y})\right] \right\} + \mathrm{const}, \tag{14}$$

where $s(\mathbf{y})$ denotes $\tau$-scaled logits predicted by the model such that $p_\theta(\mathbf{y} \mid \mathbf{x}) = f^*(s(\mathbf{y})/\tau)$, and $r(\mathbf{y}) = r(\mathbf{y}, \mathbf{y}^*)$. The gap between regularized expected reward (5) and $\tau$-scaled RML criterion (8) is simply a difference of two variances, whose magnitude decreases with increasing regularization. Proposition 2 also shows an opportunity for learning algorithms: if $\tau$ is chosen so that $q = f^*(r/\tau)$, then $f^*(a/\tau)$ and $f^*(b/\tau)$ have lower variance than $p$ (which can always be achieved for sufficiently small $\tau$ provided $p$ is not deterministic), then the expected regularized reward under $p$, and its gradient for training, can be exactly estimated, in principle, by including the extra variance terms and sampling from more focused distributions than $p$. Although we have not yet incorporated approximations to the additional variance terms into RML, this is an interesting research direction.

## 4 Related Work

The literature on structure output prediction is vast, falling into three broad categories: (a) supervised learning approaches that ignore task reward and use supervision; (b) reinforcement learning approaches that use only task reward and ignore supervision; and (c) hybrid approaches that attempt to exploit both supervision and task reward. This paper clearly falls in category (c).

Work in category (a) includes classical conditional random fields [17] and conditional log-likelihood training of RNNs [29, 2]. It also includes the approaches that attempt to perturb the training inputs

and supervised training structures to improves the robustness (and hopefully the generalization) of the conditional models (*e.g.* see [4, 16]). These approaches offer improvements to standard maximum likelihood estimation, but they are fundamentally limited by not incorporating a task reward.

By contrast, work in category (b) includes reinforcement learning approaches that only consider task reward and do not use any other supervision. Beyond the traditional reinforcement learning approaches, such as policy gradient [37, 31], and actor-critic [30], Q-learning [34], this category includes SEARN [8]. There is some relationship to the work presented here and work on relative entropy policy search [23], and policy optimization via expectation maximization [35] and KL-divergence [14, 33], however none of these bridge the gap between the two directions of the KL-divergence, nor do they consider any supervision data as we do here.

There is also a substantial body of related work in category (c), which considers how to exploit supervision information while training with a task reward metric. A canonical example is large margin structured prediction [32, 11], which explicitly uses supervision and considers an upper bound surrogate for task loss. This approach requires loss augmented inference that cannot be efficiently achieved for general task losses. We are not aware of successful large-margin methods for neural sequence prediction, but a related approach by [39] for neural machine translation builds on SEARN [8]. Some form of inference during training is still needed, and the characteristics of the objective are not well studied. We also mentioned the work on maximizing task reward by bootstrapping from a maximum likelihood policy [25, 27], but such an approach only makes limited use of supervision. Some work in robotics has considered exploiting supervision as a means to provide indirect sampling guidance to improve policy search methods that maximize task reward [18, 19, 26], but these approaches do not make use of maximum likelihood training. An interesting work is [15] which explicitly incorporates supervision in the policy evaluation phase of a policy iteration procedure that otherwise seeks to maximize task reward. However, this approach only considers a greedy policy form that does not lend itself to being represented as a deep RNN, and has not been applied to structured output prediction. Most relevant are ideas for improving approximate maximum likelihood training for intractable models by passing the gradient calculation through an approximate inference procedure [10, 28]. These works, however, are specialized to particular approximate inference procedures, and, by directly targeting expected reward, are subject to the variance problems that motivated this work.

One advantage of the RML framework is its computational efficiency at training time. By contrast, RL and scheduled sampling [4] require sampling from the model, which can slow down the gradient computation by $2\times$. Structural SVM requires loss-augmented inference which is often more expensive than sampling from the model. Our framework only requires sampling from a fixed exponentated payoff distribution, which can be thought as a form of input pre-processing. This pre-processing can be parallelized by model training by having a thread handling loading the data and augmentation.

Recently, we were informed of the unpublished work of Volkovs *et al.* [36] that also proposes an objective like RML, albeit with a different derivation. No theoretical relation was established to entropy regularized RL, nor was the method applied to neural nets for sequences, but large gains were reported over several baselines applying the technique to ranking problems with CRFs.

## 5 Experiments

We compare our approach, reward augmented maximum likelihood (*RML*), with standard maximum likelihood (*ML*) training on sequence prediction tasks using state-of-the-art attention-based recurrent neural networks [29, 2]. Our experiments demonstrate that the RML approach considerably outperforms ML baseline on both speech recognition and machine translation tasks.

### 5.1 Speech recognition

For experiments on speech recognition, we use the TIMIT dataset; a standard benchmark for clean phone recognition. This dataset consists of recordings from different speakers reading ten phonetically rich sentences covering major dialects of American English. We use the standard train / dev / test splits suggested by the Kaldi toolkit [24].

As the sequence prediction model, we use an attention-based encoder-decoder recurrent model of [5] with three 256-dimensional LSTM layers for encoding and one 256-dimensional LSTM layer for decoding. We do not modify the neural network architecture or its gradient computation in any way,

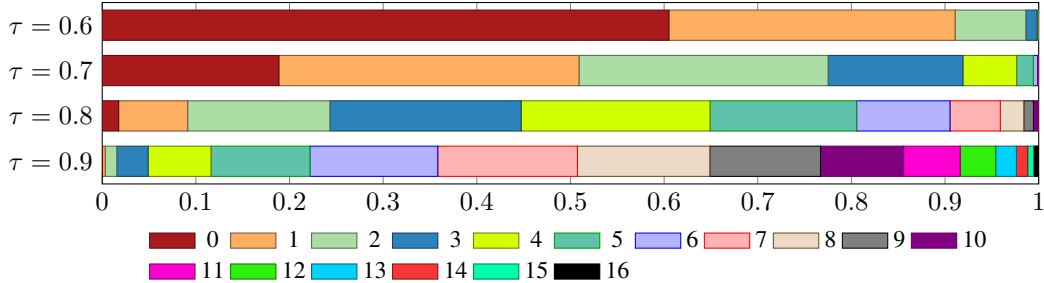

Figure 1: Fraction of different number of edits applied to a sequence of length 20 for different $\tau$. At $\tau = 0.9$, augmentations with 5 to 9 edits are sampled with a probability $> 0.1$. [view in color]

| Method | Dev set | Test set |
|---|---|---|
| ML baseline | 20.87 $(-0.2, +0.3)$ | 22.18 $(-0.4, +0.2)$ |
| RML, $\tau = 0.60$ | 19.92 $(-0.6, +0.3)$ | 21.65 $(-0.5, +0.4)$ |
| RML, $\tau = 0.65$ | 19.64 $(-0.2, +0.5)$ | 21.28 $(-0.6, +0.4)$ |
| RML, $\tau = 0.70$ | 18.97 $(-0.1, +0.1)$ | 21.28 $(-0.5, +0.4)$ |
| RML, $\tau = 0.75$ | 18.44 $(-0.4, +0.4)$ | 20.15 $(-0.4, +0.4)$ |
| RML, $\tau = 0.80$ | 18.27 $(-0.2, +0.1)$ | 19.97 $(-0.1, +0.2)$ |
| RML, $\tau = 0.85$ | 18.10 $(-0.4, +0.3)$ | 19.97 $(-0.3, +0.2)$ |
| **RML, $\tau = 0.90$** | **18.00** $(-0.4, +0.3)$ | **19.89** $(-0.4, +0.7)$ |
| RML, $\tau = 0.95$ | 18.46 $(-0.1, +0.1)$ | 20.12 $(-0.2, +0.1)$ |
| RML, $\tau = 1.00$ | 18.78 $(-0.6, +0.8)$ | 20.41 $(-0.2, +0.5)$ |

Table 1: Phone error rates (PER) for different methods on TIMIT dev and test sets. Average PER of 4 independent training runs is reported.

but we only change the output targets fed into the network for gradient computation and SGD update. The input to the network is a standard sequence of 123-dimensional log-mel filter response statistics. Given each input, we generate new outputs around ground truth targets by sampling according to the exponentiated payoff distribution. We use negative edit distance as the measure of reward. Our output augmentation process allows insertions, deletions, and substitutions.

An important hyper-parameter in our framework is the temperature parameter, $\tau$, controlling the degree of output augmentation. We investigate the impact of this hyper-parameter and report results for $\tau$ selected from a candidate set of $\tau \in \{0.6, 0.65, 0.7, 0.75, 0.8, 0.85, 0.9, 0.95, 1.0\}$. At a temperature of $\tau = 0$, outputs are not augmented at all, but as $\tau$ increases, more augmentation is generated. Figure 1 depicts the fraction of different numbers of edits applied to a sequence of length 20 for different values of $\tau$. These edits typically include very small number of deletions, and roughly equal number of insertions and substitutions. For insertions and substitutions we uniformly sample elements from a vocabulary of 61 phones. According to Figure 1, at $\tau = 0.6$, more than 60% of the outputs remain intact, while at $\tau = 0.9$, almost all target outputs are being augmented with 5 to 9 edits being sampled with a probability larger than 0.1. We note that the augmentation becomes more severe as the outputs get longer.

The phone error rates (PER) on both dev and test sets for different values of $\tau$ and the ML baseline are reported in Table 1. Each model is trained and tested 4 times, using different random seeds. In Table 1, we report average PER across the runs, and in parenthesis the difference of average error to minimum and maximum error. We observe that a temperature of $\tau = 0.9$ provides the best results, outperforming the ML baseline by 2.9% PER on the dev set and 2.3% PER on the test set. The results consistently improve when the temperature increases from 0.6 to 0.9, and they get worse beyond $\tau = 0.9$. It is surprising to us that not only the model trains with such a large amount of augmentation at $\tau = 0.9$, but also it significantly improves upon the baseline. Finally, we note that previous work [6, 7] suggests several refinements to improve sequence to sequence models on TIMIT by adding noise to the weights and using more focused forward-moving attention mechanism. While these refinements are interesting and they could be combined with the RML framework, in this work, we do not implement such refinements, and focus specifically on a fair comparison between the ML baseline and the RML method.

| Method | Average BLEU | Best BLEU |
|---|---|---|
| ML baseline | 36.50 | 36.87 |
| RML, $\tau = 0.75$ | 36.62 | 36.91 |
| RML, $\tau = 0.80$ | 36.80 | 37.11 |
| **RML, $\boldsymbol{\tau = 0.85}$** | **36.91** | **37.23** |
| RML, $\tau = 0.90$ | 36.69 | 37.07 |
| RML, $\tau = 0.95$ | 36.57 | 36.94 |

Table 2: Tokenized BLEU score on WMT'14 English to French evaluated on newstest-2014 set. The RML approach with different $\tau$ considerably improves upon the maximum likelihood baseline.

## 5.2 Machine translation

We evaluate the effectiveness of the proposed approach on WMT'14 English to French machine translation benchmark. Translation quality is assessed using *tokenized BLEU* score, to be consistent with previous work on neural machine translation [29, 2, 22]. Models are trained on the full 36M sentence pairs from WMT'14 training set, and evaluated on 3003 sentence pairs from newstest-2014 test set. To keep the sampling process efficient and simple on such a large corpus, we augment the output sentences only based on Hamming distance (*i.e.* edit distance without insertion or deletion). For each sentece we sample a single output at each step. One can consider insertions and deletions or sampling according to exponentiated sentence BLEU scores, but we leave that to future work.

As the conditional sequence prediction model, we use an attention-based encoder-decoder recurrent neural network similar to [2], but we use multi-layer encoder and decoder networks consisting of three layers of 1024 LSTM cells. As suggested by [2], for computing the softmax attention vectors, we use a feedforward neural network with 1024 hidden units, which operates on the last encoder and the first decoder layers. In all of the experiments, we keep the network architecture and the hyper-parameters fixed. All of the models achieve their peak performance after about 4 epochs of training, once we anneal the learning rates. To reduce the noise in the BLEU score evaluation, we report both peak BLEU score and BLEU score averaged among about 70 evaluations of the model while doing the fifth epoch of training. We perform beam search decoding with a beam size of 8.

Table 2 summarizes our experimental results on WMT'14. We note that our ML translation baseline is quite strong, if not the best among neural machine translation models [29, 2, 22], achieving very competitive performance for a single model. Even given such a strong baseline, the RML approach consistently improves the results. Our best model with a temperature $\tau = 0.85$ improves average BLEU by 0.4, and best BLEU by 0.35 points, which is a considerable improvement. Again we observe that as we increase the amount of augmentation from $\tau = 0.75$ to $\tau = 0.85$ the results consistently get better, and then they start to get worse with more augmentation.

**Details.** We train the models using asynchronous SGD with 12 replicas without momentum. We use mini-batches of size 128. We initially use a learning rate of 0.5, which we then exponentially decay to 0.05 after 800$K$ steps. We keep evaluating the models between 1.1 and 1.3 million steps and report average and peak BLEU scores in Table 2. We use a vocabulary 200$K$ words for the source language and 80$K$ for the target language. We only consider training sentences that are up to 80 tokens. We replace rare words with several UNK tokens based on their first and last characters. At inference time, we replace UNK tokens in the output sentences by copying source words according to largest attention activations as suggested by [22].

## 6 Conclusion

We present a learning algorithm for structured output prediction that generalizes maximum likelihood training by enabling direct optimization of a task reward metric. Our method is computationally efficient and simple to implement. It only requires augmentation of the output targets used within a log-likelihood objective. We show how using augmented outputs sampled according to edit distance improves a maximum likelihood baseline by a considerable margin, on both machine translation and speech recognition tasks. We believe this framework is applicable to a wide range of probabilistic models with arbitrary reward functions. In the future, we intend to explore the applicability of this framework to other probabilistic models on tasks with more complicated evaluation metrics.

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
