[Supplementary Material · supplement.pdf]

# Supp. Material: Reward Augmented Maximum Likelihood for Neural Structured Prediction

Mohammad Norouzi     Samy Bengio     Zhifeng Chen     Navdeep Jaitly

Mike Schuster     Yonghui Wu     Dale Schuurmans

{mnorouzi, bengio, zhifengc, ndjaitly}@google.com
{schuster, yonghui, schuurmans}@google.com

Google Brain

## A   Proofs

**Proposition 1.** *For any twice differentiable strictly convex closed potential $F$, and $p, q \in \mathrm{int}(\mathcal{F})$:*

$$D_F(q \parallel p) = D_F(p \parallel q) + \tfrac{1}{4}(p-q)^{\mathsf{T}}\big(H_F(a) - H_F(b)\big)(p-q) \qquad (1)$$

*for some $a = (1-\alpha)p + \alpha q$, $(0 \leq \alpha \leq \tfrac{1}{2})$, $b = (1-\beta)q + \beta p$, $(0 \leq \beta \leq \tfrac{1}{2})$.*

*Proof.* Let $f(p)$ denote $\nabla F(p)$ and consider the midpoint $\tfrac{q+p}{2}$. One can express $F(\tfrac{q+p}{2})$ by two Taylor expansions around $p$ and $q$. By Taylor's theorem there is an $a = (1-\alpha)p + \alpha q$ for $0 \leq \alpha \leq \tfrac{1}{2}$ and $b = \beta p + (1-\beta)q$ for $0 \leq \beta \leq \tfrac{1}{2}$ such that

$$
\begin{aligned}
F(\tfrac{q+p}{2}) &= F(p) + (\tfrac{q+p}{2} - p)^{\mathsf{T}} f(p) + \tfrac{1}{2}(\tfrac{q+p}{2} - p)^{\mathsf{T}} H_F(a)(\tfrac{q+p}{2} - p) && (2)\\
&= F(q) + (\tfrac{q+p}{2} - q)^{\mathsf{T}} f(q) + \tfrac{1}{2}(\tfrac{q+p}{2} - q)^{\mathsf{T}} H_F(b)(\tfrac{q+p}{2} - q), && (3)
\end{aligned}
$$

$$
\begin{aligned}
\text{hence,} \quad 2F(\tfrac{q+p}{2}) &= 2F(p) + (q-p)^{\mathsf{T}} f(p) + \tfrac{1}{4}(q-p)^{\mathsf{T}} H_F(a)(q-p) && (4)\\
&= 2F(q) + (p-q)^{\mathsf{T}} f(q) + \tfrac{1}{4}(p-q)^{\mathsf{T}} H_F(b)(p-q). && (5)
\end{aligned}
$$

Therefore,

$$
\begin{aligned}
F(p) + F(q) - 2F(\tfrac{q+p}{2}) &= F(p) - F(q) - (p-q)^{\mathsf{T}} f(q) - \tfrac{1}{4}(p-q)^{\mathsf{T}} H_F(b)(p-q) && (6)\\
&= F(q) - F(p) - (q-p)^{\mathsf{T}} f(p) - \tfrac{1}{4}(q-p)^{\mathsf{T}} H_F(a)(q-p) && (7)\\
&= D_F(p \parallel q) - \tfrac{1}{4}(p-q)^{\mathsf{T}} H_F(b)(p-q) && (8)\\
&= D_F(q \parallel p) - \tfrac{1}{4}(q-p)^{\mathsf{T}} H_F(a)(q-p), && (9)
\end{aligned}
$$

leading to the result. □

For the proof of Proposition 2, we first need to introduce a few definitions and background results. A Bregman divergence is defined from a strictly convex, differentiable, closed potential function $F : \mathcal{F} \to \mathbb{R}$, whose strictly convex conjugate $F^* : \mathcal{F}^* \to \mathbb{R}$ is given by $F^*(r) = \sup_{r \in \mathcal{F}}\langle r, q \rangle - F(q)$ [1]. Each of these potential functions have corresponding transfers, $f : \mathcal{F} \to \mathcal{F}^*$ and $f^* : \mathcal{F}^* \to \mathcal{F}$, given by the respective gradient maps $f = \nabla F$ and $f^* = \nabla F^*$. A key property is that $f^* = f^{-1}$ [1], which allows one to associate each object $q \in \mathcal{F}$ with its transferred image $r = f(q) \in \mathcal{F}^*$ and vice versa. The main property of Bregman divergences we exploit is that a divergence between any two domain objects can always be equivalently expressed as a divergence between their transferred images; that is, for any $p \in \mathcal{F}$ and $q \in \mathcal{F}$, one has [1]:

$$
\begin{aligned}
D_F(p \parallel q) &= F(p) - \langle p, r \rangle + F^*(r) = D_{F^*}(r \parallel s), && (10)\\
D_F(q \parallel p) &= F^*(s) - \langle s, q \rangle + F(q) = D_{F^*}(s \parallel r), && (11)
\end{aligned}
$$

where $s = f(p)$ and $r = f(q)$. These relations also hold if we instead chose $s \in \mathcal{F}^*$ and $r \in \mathcal{F}^*$ in the range space, and used $p = f^*(s)$ and $q = f^*(r)$. In general (10) and (11) are not equal.

Two special cases of the potential functions $F$ and $F^*$ are interesting as they give rise to KL divergences. These two cases include $F_\tau(p) = -\tau\mathbb{H}(p)$ and $F_\tau^*(s) = \tau\mathrm{lse}(s/\tau) = \tau\log\sum_y \exp(s(y)/\tau)$, where $\mathrm{lse}(\cdot)$ denotes the log-sum-exp operator. The respective gradient maps are $f_\tau(p) = \tau(\log(p) + \mathbf{1})$ and $f_\tau^*(s) = f^*(s/\tau) = \frac{1}{\sum_y \exp(s(y)/\tau)}\exp(s/\tau)$, where $f_\tau^*$ denotes the normalized exponential operator for $\frac{1}{\tau}$-scaled logits. Below, we derive $D_{F_\tau^*}(r\parallel s)$ for such $F_\tau^*$:

$$
\begin{aligned}
D_{F_\tau^*}(s\parallel r) &= F_\tau^*(s) - F_\tau^*(r) - (s-r)^\mathsf{T}\nabla F_\tau^*(r) \\
&= \tau\mathrm{lse}(s/\tau) - \tau\mathrm{lse}(r/\tau) - (s-r)^\mathsf{T} f_\tau^*(r) \\
&= -\tau\big((s/\tau - \mathrm{lse}(s/\tau)) - (r/\tau - \mathrm{lse}(r/\tau))\big)^\mathsf{T} f_\tau^*(r) \\
&= \tau f_\tau^*(r)^\mathsf{T}\big((r/\tau - \mathrm{lse}(r/\tau)) - (s/\tau - \mathrm{lse}(s/\tau))\big) \\
&= \tau f_\tau^*(r)^\mathsf{T}\big(\log f_\tau^*(r) - \log f_\tau^*(s)\big) \\
&= \tau D_{\mathrm{KL}}(f_\tau^*(r)\parallel f_\tau^*(s)) \\
&= \tau D_{\mathrm{KL}}(q\parallel p)
\end{aligned}
\tag{12}
$$

**Proposition 2.** *The KL divergence between $p$ and $q$ in two directions can be expressed as,*

$$
D_{\mathrm{KL}}(p\parallel q) = D_{\mathrm{KL}}(q\parallel p) + \tfrac{1}{4\tau^2}\mathrm{Var}_{y\sim f^*(a/\tau)}[s(y) - r(y)] - \tfrac{1}{4\tau^2}\mathrm{Var}_{y\sim f^*(b/\tau)}[s(y) - r(y)]
\tag{13}
$$

$$
< D_{\mathrm{KL}}(q\parallel p) + \tfrac{1}{\tau^2}\|s - r\|_2^2,
\tag{14}
$$

*for some* $a = (1-\alpha)s + \alpha r$, $(0\le\alpha\le\frac{1}{2})$, $b = (1-\beta)r + \beta s$, $(0\le\beta\le\frac{1}{2})$.

*Proof.* First, for the potential function $F_\tau^*(r) = \tau\mathrm{lse}(r/\tau)$ it is easy to verify that $F_\tau^*$ satisfies the conditions for Proposition 1, and

$$
H_{F_\tau^*}(a) = \tfrac{1}{\tau}(\mathrm{Diag}(f_\tau^*(a)) - f_\tau^*(a)f_\tau^*(a)^\top),
\tag{15}
$$

where $\mathrm{Diag}(\mathbf{v})$ returns a square matrix the main diagonal of which comprises a vector $\mathbf{v}$. Therefore, by Proposition 1 we obtain

$$
D_{F_\tau^*}(r\parallel s) = D_{F_\tau^*}(s\parallel r) + \tfrac{1}{4}(s-r)^\top(H_{F_\tau^*}(a) - H_{F_\tau^*}(b))(s-r),
\tag{16}
$$

for some $a = (1-\alpha)s + \alpha r$, $(0\le\alpha\le\frac{1}{2})$, $b = (1-\beta)r + \beta s$, $(0\le\beta\le\frac{1}{2})$. Note that by the specific form (15) we also have

$$
\begin{aligned}
(s-r)^\top H_{F_\tau^*}(a)(s-r) &= \tfrac{1}{\tau}(s-r)^\top\big(\mathrm{Diag}(f_\tau^*(a)) - f_\tau^*(a)f_\tau^*(a)^\top\big)(s-r) \tag{17} \\
&= \tfrac{1}{\tau}\big(E_{\mathbf{y}\sim f_\tau^*(a)}[(s(\mathbf{y}) - r(\mathbf{y}))^2] - E_{\mathbf{y}\sim f_\tau^*(a)}[s(\mathbf{y}) - r(\mathbf{y})]^2\big) \tag{18} \\
&= \tfrac{1}{\tau}\mathrm{Var}_{\mathbf{y}\sim f_\tau^*(a)}[s(\mathbf{y}) - r(\mathbf{y})], \tag{19}
\end{aligned}
$$

and $\quad (s-r)^\top H_{F_\tau^*}(b)(s-r) = \tfrac{1}{\tau}\mathrm{Var}_{\mathbf{y}\sim f_\tau^*(b)}[s(\mathbf{y}) - r(\mathbf{y})] \quad.$ $\tag{20}$

Therefore, by combining (19) and (20) with (16) we obtain

$$
D_{F_\tau^*}(r\parallel s) = D_{F_\tau^*}(s\parallel r) + \tfrac{1}{4\tau}\mathrm{Var}_{\mathbf{y}\sim f_\tau^*(a)}[s(\mathbf{y}) - r(\mathbf{y})] - \tfrac{1}{4\tau}\mathrm{Var}_{\mathbf{y}\sim f_\tau^*(b)}[s(\mathbf{y}) - r(\mathbf{y})].
\tag{21}
$$

Equality (13) then follows by applying (12) to (21).

Next, to prove the inequality in (14), let $\delta = s - r$ and observe that

$$
\begin{aligned}
D_{F_\tau^*}(r\parallel s) - D_{F_\tau^*}(s\parallel r) &= \tfrac{1}{4}\delta^\top\big(H_{F_\tau^*}(a) - H_{F_\tau^*}(b)\big)\delta \tag{22} \\
&= \tfrac{1}{4\tau}\delta^\top\mathrm{Diag}(f_\tau^*(a) - f_\tau^*(b))\delta + \tfrac{1}{4\tau}\big(\delta^\top f_\tau^*(b)\big)^2 - \tfrac{1}{4\tau}\big(\delta^\top f_\tau^*(a)\big)^2 \tag{23} \\
&\le \tfrac{1}{4\tau}\|\delta\|_2^2\|f_\tau^*(a) - f_\tau^*(b)\|_\infty + \tfrac{1}{4\tau}\|\delta\|_2^2\|f_\tau^*(b)\|_2^2 \tag{24} \\
&\le \tfrac{1}{2\tau}\|\delta\|_2^2 + \tfrac{1}{4\tau}\|\delta\|_2^2 \tag{25}
\end{aligned}
$$

since $\|f_\tau^*(a) - f_\tau^*(b)\|_\infty \le 2$ and $\|f_\tau^*(b)\|_2^2 \le \|f_\tau^*(b)\|_1^2 \le 1$. The result follows by applying (12) to (25). $\qquad\square$

## References

[1] A. Banerjee, S. Merugu, I. S. Dhillon, and J. Ghosh. Clustering with Bregman divergences. *JMLR*, 2005.