[Reviews · NeurIPS 2016]

Reviewer 1

Summary

The proposed approach consists in corrupting the training targets with a noise derived from the task reward while doing maximum likelihood training. This simple but specific smoothing of the target distribution allows to significantly boost the performance of neural structured output prediction as showcased on TIMIT phone and translation tasks. The link between this approach and RL-based expected reward maximization is also made clear by the paper,

Qualitative Assessment

The paper is a superbly written account of a simple idea that appears to work very well. The approach can straightforwardly be applied to existing max-likelihood (ML) trained models in order to in principle take into account the task reward during training and is computationally much more efficient than alternative non ML based approaches. This work risks being underappreciated as proposing but a simple addition of artificial structured-label noise, but I think the specific link with structured output task reward is sufficiently original, and the paper also uncovers important theoretical insight by revealing the formal relationship between the proposed reward augmented ML and RL-based regularized expected reward objectives. However, the kind of substitution-deletion-insertion corruption that you use in your experiments would come to mind as a natural way to augment this type of targets, to smooth the target distribution in a plausible way, even in the absence of any task reward. So while it works surprisingly well, you haven't yet clearly demonstrated empirically that using a truly *task-reward derived* payoff distribution is beneficial. One way to convincingly demonstrate that would be if you did your envisioned BLEU importance reweighted sampling, and were able to show that it improves the BLEU test score over your current simpler edit-distance based label noise. - It would be very interesting if in table 1 you also provided the resulting phone error rate on the training set (on its clean targets), to see to what degree (if any) it deteriorates with the addition of noise during training (does it reach the level of the dev/test sets?) - Typos and other suggestions for improvement: l 83: "is finite" -> why this assumption? l 86: "model's" -> "model" l 189: p,q \in \Delta^{|Y|} -> undefined notation l 192: "divided" -> shouldn't it say multiplied here (as f_\tau subsequently divides by \tau) l 198..203 This paragraph is a little too dense to easily follow. l 247: "exponentated" -> "exponentiated" l 248: "a form of input pre-processing" -> or rather target pre-processing?

Confidence in this Review

3-Expert (read the paper in detail, know the area, quite certain of my opinion)


Reviewer 2

Summary

This paper proposes a novel learning objective for training structured predictors, including non-log-linear, neural ones. Prior work has chosen either maximum likelihood learning, which is relatively tractable but assumes a log likelihood loss, or reinforcement learning, which can be performed for a task-specific loss function but requires sampling many predictions to estimate gradients. The proposed objective bridges the gap with "reward-augmented maximum likelihood," which is similar to maximum likelihood but estimates the expected loss with samples that are drawn in proportion to their distance from the ground truth. Empirical results show good improvements with LSTM-based predictors on speech recognition and machine translation benchmarks relative to maximum likelihood training.

Qualitative Assessment

This paper's motivating insight, the symmetry between maximum likelihood and reinforcement learning objectives when viewed as a variational minimization, is very insightful. It motivates replacing the delta distribution that arises in this view of maximum likelihood learning with a "softer" distribution over many possible predictions, which are selected based on how close they are to the ground truth structure as determined by task-specific loss. In some ways it seems analogous to maximum margin learning, in that learning with respect to a task-specific loss is made more tractable by putting loss terms in the objective for many different possible predictions, but as the authors point out, it is doubtful that the convex upper bound of maximum margin learning is tight for deep neural networks. The proposed approach instead augments the learning objective by sampling points in the output space in proportion to how good they are. This is a very nice idea that gives good performance boosts on the benchmark tasks considered. The paper is very well written and clear. Questions: Is it obvious that models trained with the relational learning objective wouldn't work as well or would be too expensive to compare with? The experiments consider and illustrate nicely the effects of different choices for the temperature parameter \tau, but Is the choice of the number of samples used to augment the objective a separate hyperparameter whose choice is orthogonal? I don't see a discussion of how that was tuned. I'd be curious to know it affects both accuracy and computational cost.

Confidence in this Review

2-Confident (read it all; understood it all reasonably well)


Reviewer 3

Summary

The papers suggests a new approach to incorporate reward function (or equivalently loss function) to Conditional random fields (CRFs). This work is inspired by recent advancement in reinforcement learning and likelihood learning. The authors suggest to learn parameters so as to minimize the KL divergence between CRFs and a probability model that is proportional to the reward function (which the authors call payoff distribution, see Equation 4). The authors suggest an optimization algorithm for the KL-divergence minimization that depends on sampling from the payoff distribution.

Qualitative Assessment

The authors suggest a very interesting idea for CRF optimization. As far as I know, this is an original direction which certainly should be explored. The main technical difficulty is sampling from the payoff distribution, which should be #P-hard for general payoff functions. This should be hard since the payoff function may have no structure (e.g. edit-distance or BLEU score). The authors suggest a heuristic to sample from such distributions. While it is an interesting direction, I am not sure it is fully developed. Lastly, as the paper's idea is separate from deep learning, I am not sure that connecting it to deep learning architectures helps to clarify its ideas.

Confidence in this Review

2-Confident (read it all; understood it all reasonably well)


Reviewer 4

Summary

Current methods to learn a model for structured prediction include max margin optimisation and reinforcement learning. However, the max margin approach only optimises a bound on the true reward, and requires loss augmented inference to obtain gradients, which can be expensive. On the other hand, reinforcement learning does not make use of available supervision, and can therefore struggle when the reward is sparse, and furthermore the gradients can have high variance. The paper proposes a novel approach to learning for problems that involve structured prediction. They relate their approach to simple maximum likelihood (ML) learning and reinforcement learning (RL): ML optimises the KL divergence of a delta distribution relative to the model distribution, and RL optimises the KL divergence of the model distribution relative to the exponentiated reward distribution. They propose reward-augmented maximum likelihood learning, which optimises the KL divergence of the exponentiated reward distribution relative to the model distribution. Compared to RL, the arguments of the KL divergence are swapped. Compared to ML, the delta distribution is generalised to the exponentiated reward distribution. Training is cheap in RML learning. It is only necessary to sample from the output set according to the exponentiated reward distribution. All experiments are performed in speech recognition and machine translation, where the structure over the output set is defined by the edit distance. An improvement is demonstrated over simple ML.

Qualitative Assessment

This is an elegant solution to a clear problem in machine learning: how to efficiently and exactly optimise the reward when the output is chosen by the discontinuous argmax function and/or the error metric is discontinuous (for example the 0-1 loss). I found the paper to be extremely clear. The authors frequently gave succinct and useful definitions of the problems they face and solution they propose. It would be good to see a comparison to reinforcement learning in experiments if possible? Or, if it's difficult to obtain this, then state that? I'm excited to see this approach applied to object detection problems. Do you think it's feasible?

Confidence in this Review

2-Confident (read it all; understood it all reasonably well)


Reviewer 5

Summary

The submitted paper presents an alternative to expected reward optimization (eq. 3) for neural structured prediction. The new objective (eq. 8) requires to work with a normalized reward score that defines a proper probability distribution. Sampling from this stationary distribution has the great advantage of reduced variance (which is furthered by annealing with a temperature hyperparameter). The paper is evaluated on two sequence-to-sequence learning tasks, with promising results.

Qualitative Assessment

The paper addresses an important problem in optimizing expected reward for structured prediction, namely the high variance of estimation procedures that have to sample from a non-stationary model distribution. In the natural language community, especially in statistical machine translation, "expected reward optimization" is known under the name of "minimum risk optimization" or "maximum expected BLEU" training, and has been successfully applied for linear discriminative models (Och: Minimum error rate training in statistical machine translation, ACL 2003; Li & Eisner: First- and second-order expectation semirings with applications to minimum-risk training on translation forests, EMNLP 2009; He & Deng: Maximum expected BLEU training of phrase and lexicon translation models; inter alia) and non-linear neural models (Shen et al.: Minimum risk training for neural machine translation, ACL 2016). It is also known in vision as "expected loss optimization" (Yuille & He: Probabilistic models of vision and max-margin methods. Frontiers of Electrical and Electronic Engineering. 2012). In all these approaches, instead of sampling, expected reward is calculated exactly on n-best lists of outputs or via dynamic programming over the output graph. These approaches should be mentioned, especially the application to neural MT by Shen et al. 2016, and it should be explained why a direct calculation of expected reward over n-best lists or translation graphs is not an option in your approach. Additionally, since these approaches are well-known in machine translation, they should be considered as baseline in your experimental comparison. The issue of sampling becomes important if reward feedback on output is only available for a single sampled output. This is the case in Ranzato et al.'s reinforcement learning approach or in bandit structure prediction approaches (Sokolov et al.: Stochastic Structured Prediction under Bandit Feedback. http://arxiv.org/abs/1606.00739). The first approach solves the variance problem by initializing the model with a word-level model trained in batch-mode. The second approach explores different objectives among which a cross-entropy objective is used which is identical to (7) (or (8) without the maximum entropy regularizer). Since the approach of Solokov et al. works in a bandit setting, normalizing reward in order to sample from it is not possible. Thus one drawback of the presented work is that is applicable only in cases where rewards are available to all output structures, otherwise the normalization constant cannot be calculated properly. Speaking of which, what are the guarantees on the quality of approximating the normalization constant based on samples? Firstly, a normalization constant is not an expectation that can be easily be approximated by sampling. Secondly, an approximate per-sentence BLEU score needs to be used in the practical application to machine translation. What is it in your experiment? Concerning the analysis given in the paper, it is nice to see how objective (8) and (3) differ. However, a simple application of Jensen's inequality (mentioned in Sokolov et al. 2016) shows that (8) is a convex upper bound on (3): Let q and p be defined as in Section 3, then E_q[ -log p] >= -log E_p[q]. It is not entirely clear what the lengthy derivation given in the presented paper adds to this. Furthermore, for unbounded SMT metrics such as Translation Error Rate (Snover et al.: A Study of Translation Error Rate with Targeted Human Annotation, AMTA 2006) the term ||s-r|| can be very large. Thus r and s need to be specified more clearly in the submitted paper. What is also not mentioned in the presented paper is the aspect of convexity which could give rise to faster convergence, which is again not mentioned or measured in the presented paper. Since variance reduction is claimed as a key contribution, the relation of variance reduction to convergence speed might be worth exploring. Lastly, the experiments presented in the paper seem to show promising improvements, however, many details are missing: What was the per-sentence BLEU approximation used as reward signal? Why did you not optimize the temperature constant on a heldout set - the presented evaluation basically selects the temperature constant on the test set. Are the result differences statistically significant? Finally, possible citations for using annealing schedules and sampling in expected loss training for structured prediction are: Smith & Eisner: Minimum risk annealing for training log-linear models. ACL 2006. Arun et al.: A unified approach to minimum risk training and decoding. WMT 2010.

Confidence in this Review

2-Confident (read it all; understood it all reasonably well)


Reviewer 6

Summary

This paper presents a novel extension of maximum likelihood training for structured prediction problems. The proposed objective (reward-augmented maximum likelihood, abbreviated as RML) is defined as the KL divergence between a prior of outputs (in terms of the reward/loss function, so called "exponentiated payoff distribution" given at line 104) and the model density. As a special case, the maximum likelihood (ML) training takes a delta distribution as the prior without considering the reward function. On the other hand, the objective of reinforcement learning (RL) has a similar form except that it uses a KL divergence in the opposite direction. The connection between RML and RL is analyzed in terms of Bregman divergence. Experiments are conducted for speech recognition and machine translation.

Qualitative Assessment

Novelty and potential impact or usefulness: The paper provides a nice unified point of view for several learning objectives in terms of the KL divergence. From computational and statistical perspectives, RML is a good combination of ML and RL. However, there is a common obstacle for all structured prediction methods, that is, the integral over the output space is intractable. Unfortunately, to optimize a RML problem, sampling cannot be avoid in general, although it is more efficient than that of RL. ML can be viewed as pushing the model distribution towards the delta distribution. Given the samples, RML leads to a model distribution resembling a weighted sum of delta distributions. From the computational perspective, RML can be viewed as ML with manually created samples, thus more costly. But on the other hand, thanks to GPU parallel computing, a deep model may benefit from these additional training samples. For loss-augmented learning, it is questionable which method is the best, since loss-augmented inference could be as efficient as sampling given recent development of approximate MAP inference (e.g., [*]). It is also possible to use block coordinate descent without performing global inference [**]. [*] Meshi et al. Smooth and Strong: MAP Inference with Linear Convergence. NIPS 2015. [**] Meshi et al. Efficient Training of Structured SVMs via Soft Constraints. AISTATS 2015. Technical quality: > The method is well presented. Proofs are sound. However, the analysis in section 3 is not reflected in the experiments to demonstrate the effectiveness of RML. > Experimental issues: - There are several important baselines are missing: RL, structured SVMs and loss-augmented CRF, which are mentioned in introduction and related work. - How many y's are sampled for each experiment? Are these number critical to the experiments? Clarity and presentation: The paper is well written. However the title "neural structured prediction" may be misleading since the method is not only for deep models.

Confidence in this Review

2-Confident (read it all; understood it all reasonably well)